# Epimers of Vitamin D: A Review

**DOI:** 10.3390/ijms21020470

**Published:** 2020-01-11

**Authors:** Bashar Al-Zohily, Asma Al-Menhali, Salah Gariballa, Afrozul Haq, Iltaf Shah

**Affiliations:** 1Department of Chemistry, College of Science, United Arab Emirates University, Al Ain 15551, UAE; 201250397@uaeu.ac.ae; 2Department of Biology, College of Science, United Arab Emirates University, Al Ain 15551, UAE; 3Internal Medicine, Faculty of Medicine & Health Sciences, United Arab Emirates University, Al Ain 15551, UAE; s.gariballa@uaeu.ac.ae; 4Department of Food Technology, School of Interdisciplinary Sciences and Technology, Jamia Hamdard University, New Delhi-110062, India; haq2000@gmail.com

**Keywords:** epimers of vitamin d, 3-epi-25OHD3, vitamin D, 25OHD, 25OHD3, C3 epimer

## Abstract

In this review, we discuss the sources, formation, metabolism, function, biological activity, and potency of C3-epimers (epimers of vitamin D). We also determine the role of epimerase in vitamin D-binding protein (DBP) and vitamin D receptors (VDR) according to different subcellular localizations. The importance of C3 epimerization and the metabolic pathway of vitamin D at the hydroxyl group have recently been recognized. Here, the hydroxyl group at the C3 position is orientated differently from the alpha to beta orientation in space. However, the details of this epimerization pathway are not yet clearly understood. Even the gene encoding for the enzyme involved in epimerization has not yet been identified. Many published research articles have illustrated the biological activity of C3 epimeric metabolites using an in vitro model, but the studies on in vivo models are substantially inadequate. The metabolic stability of 3-epi-1α,25(OH)_2_D3 has been demonstrated to be higher than its primary metabolites. 3-epi-1 alpha, 25 dihydroxyvitamin D3 (3-epi-1α,25(OH)_2_D3) is thought to have fewer calcemic effects than non-epimeric forms of vitamin D. Some researchers have observed a larger proportion of total vitamin D as C3-epimers in infants than in adults. Insufficient levels of vitamin D were found in mothers and their newborns when the epimers were not included in the measurement of vitamin D. Oral supplementation of vitamin D has also been found to potentially cause increased production of epimers in mice but not humans. Moreover, routine vitamin D blood tests for healthy adults will not be significantly affected by epimeric interference using LC–MS/MS assays. Recent genetic models also show that the genetic determinants and the potential factors of C3-epimers differ from those of non-C3-epimers.Most commercial immunoassays techniques can lead to inaccurate vitamin D results due to epimeric interference, especially in infants and pregnant women. It is also known that the LC–MS/MS technique can chromatographically separate epimeric and isobaric interference and detect vitamin D metabolites sensitively and accurately. Unfortunately, many labs around the world do not take into account the interference caused by epimers. In this review, various methods and techniques for the analysis of C3-epimers are also discussed. The authors believe that C3-epimers may have an important role to play in clinical research, and further research is warranted.


**Highlights**


All major vitamin D forms can undergo epimerization at the C3 position, leading to the formation of C3-epimers that can overestimate vitamin D status in routine laboratory tests.Higher levels of C3-epimers are observed in mothers and newborns.Oral supplementation of vitamin D can cause an increased production of epimers in mice.LC-MS/MS can be used to separate C3-epimers from other vitamin D metabolites.

## 1. Introduction

Recently it was found that C3-epimers of vitamin D may have an important role to play in clinical research. The exact source of C3-epimers is not known but it was found that oral supplementation of vitamin D can cause increased production of epimers in mice but not in humans [1]. Moreover, it is known that all major vitamin D metabolites can be epimerized at the C3 position, with higher amounts in infants [2]. Furthermore, mothers and newborns are known to have high levels of C3-epimers [3]. Recent genetic models also show that the genetic determinants and potential factors of C3-epimers differ from those of non-C3-epimers [4]. It is also of importance that the C3-epimers can cause an overestimation of vitamin D status in routine laboratory tests [5]. However, very few labs in the world take into account that measurements can be misleading due to overlapping C3-epimers and that these co-eluting C3-epimers can be separated from vitamin D metabolites using LC-MS/MS techniques [6]. Studies have shown that routine vitamin D blood tests for healthy adults are not significantly affected by epimeric interference using LC-MS/MS assays [7].

Vitamin D is a prohormone; it is fat-soluble and derived from cholesterol. Vitamin D is obtained from two sources: the first source is sunlight or artificial ultraviolet-B (UV-B) exposure and the second source is from food like fish, eggs, milk, and cereals, or vitamin D supplements [8,9]. Exposure to UVB is a major source of vitamin D, and limited exposure to UVB radiation can negatively impact vitamin D synthesis in the skin [10]. It is also known that people with darker skin (abundant melanin) require more prolonged sun exposure compared to light-skinned people [11]. In some countries in the world, despite the year-round sunlight, there is a high prevalence of vitamin D deficiency [10,12,13]. Vitamin D deficiency is correlated with the development of many diseases, including rickets, osteomalacia, arthritis, diabetes, dementia, Parkinson’s, Alzheimer’s, and cardiovascular diseases [5]. Furthermore, vitamin D deficiency is also associated with aging, obesity, skeletal muscle weaknesses, and metabolic syndrome diseases. The role of epimers in disease onset and progression is not clearly understood, but it is known that epimers have a potent role as compared to their corresponding non-epimeric forms [9,10,14,15,16,17].

There are two primary metabolites of vitamin D, vitamin D3 and vitamin D2, which are collectively known as vitamin D [5,18]. Vitamin D3 is formed from its precursor 7-dehydrocholesterol in the skin by ultraviolet B light (medium wavelength, 290–315 nm). In the first step, 7-dehydrocholesterol is converted to pre-vitamin D3 which is followed by conversion of pre-vitamin D3 to vitamin D3, as shown in Figure 1 [19,20,21]. The second step is governed by the conversion of vitamin D3 to 25OHD3 in the liver via the 25-hydroxylase (CYP2R1) enzyme, as shown in Figure 1 [22]. In the liver, the 3-epimerase enzyme also converts 25OHD3 to 3-epi-25OHD3. 25OHD3 and 3-epi-25OHD3 are converted in the kidneys into 1α,25(OH)_2_D3 via the action of the enzyme 1α-hydroxylase (CYP27B1), as shown in Figure 1 [23,24]; 3-epimerse enzymes also convert 25OHD3 into 3-epi-1α,25(OH)_2_D3. Moreover, 3-epi-25OHD3 also becomes converted into 3-epi-1α,25(OH)_2_D3 in the kidneys. Likewise, 1α,25(OH)_2_D3 is converted into 3-epi-1α,25(OH)_2_D3 via the action of the epimerase enzyme. Sometimes, however, 25OHD3 and 1α,25(OH)_2_D3 undergo further oxidation using enzyme epimerases and are converted into epimeric forms like 3-epi-25OHD3, 3-epi-1α,25(OH)_2_D3, and 3-epi-24R,25(OH)_2_D3.

In neonatal human keratinocytes, Reddy and colleagues have concluded that 3-epi-1α,25(OH)_2_D3 is metabolized through the C-24 oxidation pathway to produce three polar compounds: 3-epi-1α,24,25(OH)_3_D3, 3-epi-24-oxo-1α,25(OH)_2_D3, and 3-epi-24-oxo-1α,23(S),25(OH)_3_D3, while C-23 oxidation of 3-epi-1α,25(OH)_2_D3 produces 3-epi-1α,23(S),25(OH)_3_D3, as shown in Figure 2 [19,25]. The C-24 oxidation pathway is catalyzed by the 1α,25(OH)_2_D3-24-hydroxylase (CYP24) enzyme. The biological activity of 3-epi-1α,25(OH)_2_D3 and its metabolites have not yet been identified [25].

In the epimerization pathway, for example, the intermediate metabolites of major vitamin D precursors are converted to epimers which undergo the same hydroxylation and oxidation events by the same enzymes when compared to the standard metabolic pathway, which also subsequently leads to the production of epimers like 3-epi-25OHD3, 3-epi-1α,25(OH)_2_D3, and 3-epi-24(R)25(OH)_2_D3 [19,26,27,28,29,30].

To write this review, we searched electronic databases, including NCBI PubMed, Google Scholar, SciFinder, and ScienceDirect, for English language articles. This was done using keywords such as epimers of vitamin D, C3-epimers, metabolism of vitamin D epimers, epimerization pathway for standard metabolism of vitamin D, function of the epimer of 1α,25(OH)_2_D, detection and quantitative analysis of vitamin D epimers, and the effects of epimers on routine analysis of total circulating vitamin D (25OHD) and its determination in serum. Due to an increased interest in epimers of vitamin D and their possible relationship with health and disease, we decided to carry out this fascinating literature survey.

## 2. C3-Epimerization of Vitamin D

The epimers of 25OHD3 and 1α,25(OH)_2_D3 were also found in rats when given pharmacological doses of 25OHD3 and 1α,25(OH)_2_D3 supplements over a period of time [26,30]. The epimerase enzyme (25OHD3-3-epimerase) is known to be responsible for the epimerization of 25OHD3 at the C3 location in the endoplasmic reticulum of liver, bone, and skin cells. However, the gene responsible for encoding the epimerization enzyme has not yet been identified [23]. NADPH is also utilized as a cofactor for the action of 25OHD3-3-epimerase in the microsomes of osteoblastic UMR-106 cells (these are transplantable rat osteogenic sarcoma cell lines, and these cell lines are responsive to PTH, prostaglandins, and bone-resorbing steroids like vitamin D). Furthermore, epimerase enzymes can carry out the epimerization process of 1α,25(OH)_2_D3 and 24(R)25(OH)_2_D3 but not at the same rate as 25OHD3. Moreover, the observation shows that this process is irreversible [23,31]. 

It is known that the epimerization process for 25OHD3, 1α,25(OH)_2_D3, and 24(R)25(OH)_2_D3 occurs in some specific culture cells, including quiescent human colon adenocarcinoma cells (Caco-2 colon carcinoma cells), human hepatoblastoma cells (HepG2 cells), human osteosarcoma (MG-63) [32], bovine parathyroid cells [29], and porcine kidney cells (LLC-PK1) [32]. Vitamin D supplementation has recently become common, and more studies are recommended to determine the origin of epimers, either produced by exogenous sources or when metabolized endogenously; additionally, it will be beneficial to understand the nature and properties of the enzymes or tissues responsible for the epimerization process to obtain an in-depth understanding of epimers [19]. A recent study reported that 25OHD3, 1α,25(OH)_2_D3, and 24,25(OH)_2_D3 can be respectively epimerized into 3-epi-25OHD3, 3-epi-1α,25(OH)_2_D3, and 3-epi-24,25(OH)_2_D3 in many different cell cultures types, such as LLC-PK1, Caco-2, UMR-106, Hep-G2, and MG-63 cells, but at different proportions. For example, the C3 epimerization of 25OHD3 and 1α,25(OH)_2_D3 mostly occurs in equivalent amounts in UMR-106, LLC-PK1, and Caco-2. However, compared to other epimers, Hep-G2 and MG-63 cells mostly produce 3-epi-25OHD3, while 3-epi-24,25(OH)_2_D3 is the least produced of the epimers [26].

## 3. Role of Epimers in Calcium, Phosphorus and PTH Homeostasis

As mentioned earlier, 1α,25(OH)_2_D3 is considered to be the most effective vitamin form; its principal function is to increase the amount of calcium and phosphate to normal levels in plasma, and it is also required to optimize bone health. Moreover, 1α,25(OH)_2_D3 suppresses parathyroid hormones (PTH) by binding to VDR, thereby inhibiting gene expression and cell proliferation and causing the levels of calcium to increase [8,19]. It is known that 3-epi-1α,25(OH)_2_D3 has fewer calcemic effects than non-epimeric forms of vitamin D [33] but, compared to other epimeric forms of vitamin D, it is the most efficient [26]. It was also recently discovered that 3-epi-25OHD3 and its calcemic effects are higher than those of 3-epi-24,25(OH)_2_D3 [26]. 3-epi-25OHD3 and 3-epi-1α,25(OH)_2_D3 have less affinity toward DBP and even lower affinity for VDR compared to primary metabolites 25OHD3 and 1α,25(OH)_2_D3, which will lead to a reduction in the ability of epimers to induce calcium transport, as well as a much-reduced gene expression in the human colonic carcinoma cell line, Caco-2 [34,35]. For example, the bone gamma-carboxy glutamic acid-containing protein (BGLAP, osteocalcin) and the *CYP24* gene are activated by 3-epi-1α,25(OH)_2_D3 but at a much lower rate than 1α,25(OH)_2_D3 [19,26,36]. Furthermore, 3-epi-1α,25(OH)_2_D3 has less control on the antiproliferation and differentiation of cells compared to 1α,25(OH)_2_D3, such as HL-60 promyelocytic leukemia cells, keratinocytes, and rat UMR-106 osteosarcoma cells [19,23]. Moreover, parathyroid hormone secretions can be suppressed by 3-epi-1α,25(OH)_2_D3 in bovine parathyroid cells with almost the same potency as its parent metabolite [29]. Moreover, some biological studies conducted on pulmonary alveolar type II cells show that 3-epi-1α,25(OH)_2_D3 can boost the synthesis of surfactant phospholipids and activate gene expression to yield an increase in the synthesis of surfactant protein-B (SP-B) [19,33]. Also, 3-epi-1α,25(OH)_2_D3 has greater metabolic stability than 1α,25(OH)_2_D3 [19,37]. For example, the rate of metabolism (side-chain oxidation) by CYP24A1 in human keratinocytes for 3-epi-1α,25(OH)_2_D3 is less than that of its parent metabolite [23,38,39]. The biological activity of 3-epi-25OHD in humans has not yet been identified [40]. The biological activity of the C3-epimer has been demonstrated in most in vitro models; however, the physiological functions of the C3-epimer remain unclear despite having been studied using in vivo models [19,34,41].

The C3-epimer has been shown to possess some calcemic and non-calcemic regulatory effects compared to its non-epimeric form. Compared to the respective 25OHD3 and 1α,25(OH)_2_D3 forms, the C3-epimers (3-epi-25OHD3 and 3-epi-1α,25(OH)_2_D3) bind to DBP at about 36–46% and VDR at 2–3%. However, 3-epi-1α,25(OH)_2_D3 induces BLGP (osteocalcin) VDR-binding downstream at only ~15%, as well as *CYP24* gene expression, compared to 1α,25(OH)_2_D3. Similarly, 3-epi-1α,25(OH)_2_D3 has been shown to possess more differentiation or antiproliferative activities (approximately 30% and 10%) than the non-epimeric compound. It is also known that the parathyroid hormone is suppressed by 3-epi-1α,25(OH)_2_D3 and that epimers are responsible for inducing phospholipid synthesis in pulmonary alveolar type II cells in comparable amounts to the non-epimeric form. Moreover, 3-epi-1α,25(OH)_2_D3 has been proposed to have higher metabolic stability than 1α,25(OH)_2_D3, despite having non-equivalent VDR binding [19]. Due to its weak interactions with VDR, the vitamin D epimer will remain in a free form, which might affect bodily function and give false-positive results using normal measurement methods. 

## 4. Potency of Epimerization in Microsomal Fractions

Bone cells (UMR-106) were used to collect subcellular fractions (homogenate, nuclei, mitochondria, and microsome fractions), and when 1α,25(OH)_2_D3 was incubated with these fractions, the potency of epimerization was observed to be the highest in microsomal fractions compared to others. Moreover, when comparing the epimerization activity of liver cells (HUH-7, HepG2), colon cells (Caco2), and bone cells (MG-63, UMR-106), it was found that the highest proportions of epimerization were found in the microsomal fractions for UMR-106 cells [31]. Recently, it was found that 3-epi-24,25(OH)_2_D3 could be formed from the epimerization of 24,25(OH)_2_D3, and this reaction is catalyzed by 3α-HSD and β-HSD (studied in the presence of testosterone in *Pseudomonas* bacteria), where the nicotinamide adenine dinucleotide (NAD) and nicotinamide adenine dinucleotide phosphate (NADPH) act as coenzymes in the reaction [42]. It was also found that this reaction could be modified for non-epimeric forms, like 24,25(OH)_2_D3, which could also be catalyzed by the same enzymes (3α-HSD and β-HSD) [31]. Furthermore, it was noted that the cytosol is the region where β-HSD and α-HSD could be most efficient [43,44,45], and it was reported that 3α-HSD and β-HSD enzymes could also be responsible for epimerization in microsomal fractions but in very small proportions [31].

## 5. The Role of DBP, VDR, and Genetics in Epimerization

The differences in the molecular structures of 25OHD3 and its epimer lie in the configuration of only one functional group at a specific carbon (C-3), whereby the hydroxyl groups in 3-epi-25OHD3 and 3-epi-1α,25(OH)_2_D3 have different orientations in space compared to 25OHD3 and 1α,25(OH)_2_D3, respectively (C-3α vs. C-3β) [6,23]. The vitamin D-binding protein (DBP) and vitamin D receptor (VDR) are two proteins that are central to the metabolism and mechanism of action related to the circulation of the vitamin D metabolite, 1α,25(OH)_2_D3 [23,46]. The vitamin D-binding protein (DBP) transports vitamin D metabolites through blood vessels toward different tissues, and these metabolites rarely circulate in free form. The liver is one of the organs that are responsible for producing DBP. Liver, intestinal, or renal diseases will lead to a decrease in DBP, which will cause a reduction in vitamin D metabolites and produced epimers. However, this does not mean that people with low DBP are deficient in vitamin D as the free form of vitamin D could be within the normal range [46]. When active 1α,25(OH)_2_D3 reaches a target cell, it is released from the DBP. After that, it is attached to vitamin D receptors on the cells, and the target cells will uptake this free active metabolite 1α,25(OH)_2_D3 inside the cell; the metabolite will then either be rapidly metabolized by the 1α,25(OH)_2_D-24-hydroxylase (CYP24A1) enzyme through the C-24/23 oxidation pathway leading to the formation of other metabolites, or it will be re-attached to VDR [19,23,46]. The VDR will then go through conformational changes within the nucleus that will allow other transcriptional factors to combine with 1α,25(OH)_2_D3, which will ultimately influence gene transcription. To initiate gene expression, the active vitamin D–VDR complex must interact with retinoid X receptor (RXR), which recognizes the selective or promoter sites of DNA and transcription begins; however, transcriptional activity and VDR binding affinity were found to be weaker for 3-epi-1α,25(OH)_2_D3 than for 1α,25(OH)_2_D3. Therefore, it was concluded that 3-epi-1α,25(OH)_2_D3 performs gene regulation in the same manner as its non-epimeric analogue, but less efficiently. It was also concluded that epimers will have limited binding capabilities to DBP and VDR which will, in turn, influence gene transcription. This difference in orientation (C-3α vs. C-3β) likely makes this affinity and potency differ between 1α,25(OH)_2_D3 and its epimer, as shown in Figure 3 [26,46,47].

For example, VDR binds with an active vitamin D metabolite to induce genomic action which inhibits growth in renal osteodystrophy, breast, or prostate primary tumor cells, lymphomas, psoriasis, or in autoimmune diseases and osteoporosis [46,48]. However, the role of epimers in these contexts remains unclear. A recent study hypothesized that genetic polymorphisms in the vitamin D-related gene pathway cause variation in C3-epimer levels. In this study, candidate single-nucleotide polymorphisms (SNPs) with regard to C3-epimer levels were investigated. Interestingly, it was noted that participants carrying a minor T allele exhibited a tendency to increase their levels of C3-epimer, while those carrying a minor G allele tended to produce a decreased level of both non-C3-epimers and C3-epimers [4].

## 6. Vitamin D Epimer Levels in Humans and Mice

The source of vitamin D epimers, whether from endogenous metabolism, diet, or supplementation, remains unclear [49]. Previously, it was found that, if endogenously synthesized, the immaturity of renal and hepatic tissues could be responsible for epimerization of the major metabolites of vitamin D. Subsequently, a new study has found that there is no difference in the epimer levels between healthy adults and adults with compromised liver function [2,34]. Previously, it was found that supplements of vitamin D3 are not a possible source of epimerization and, therefore, do not constitute a potential cause [50]. However, a recent study has shown that epimer levels are proportionally higher with oral supplementation of vitamin D compared to the ultraviolet irradiation of mice skin. This was not the case in humans [1]. A recent study where oral supplementation was compared with sunshine exposure in an animal model vs. human model found that the total levels of epimers (3-epi-25OHD3) were higher with oral supplementation than with sunlight exposure in an animal model. Further experiments showed that sun exposure causes a decline in epimer levels in the animal model; however, the mechanism for this process was not identified. Nonetheless, it was noted that several variables might be responsible for the breakdown or production of epimers. It was found that *CYP2R1* gene expression in mice livers is greater with vitamin D supplements compared to sunshine irradiated mice [1].

It is not known whether the epimerization process happens before or after hydroxylation. The epimerization process takes place in various other tissue types, including kidney, colon, liver, and bone cells, and the degree of epimerization is different between these tissues [1,26]. Epimerization could also occur in the skin and lead to the epimerization of pre-vitamin D3, and therefore, the expression of 25-hydroxylase (CYP2R1) will be less efficient, thus leading to a decrease in 3-epi-25OHD3 production [1]. One study found a considerable increase in the expression of the *CYP24A1* gene in the renal cells of mice exposed to sunlight (UV-irradiated) compared to those with vitamin D supplements [51]. When the activity of CYP24A1 enzyme increases, the catabolism of 25OHD3, 1α,25(OH)_2_D3, and the corresponding epimeric forms increases. Thus, these epimers could also be decreased by an increase in the activity of the CYP24A1 enzyme [1]. Recently, it was found that vitamins D3 and D2 can also be hydroxylated by the CYP11A1 enzyme, which leads to the production of a few new vitamin D metabolites, 20, 22(OH)_2_D3 or D2 and 20OHD3 or D2. Thus, it was found that CYP27A1, CYP27B1, and CYP24A1 can also carry out further hydroxylation [52,53]. The activity of the CYP11A1 enzyme has been observed in epidermal keratinocytes, where 20, 22(OH)_2_D3 and 22OHD3 were formed in higher proportions [54]. If true, CYP11A1 may be activated by UVB exposure, and the availability of the vitamin D3 substrate for classical hydroxylation to form 25OHD3 and its epimer could be lower in the absence of sunlight [1]. 

Vitamin D metabolism is affected by different characteristics of humans and mice. The first characteristic is the structure of the skin. For instance, the melanocytes in human skin are present at the basal layer of the epidermis, while in mice, melanocytes are present in the hair follicle and the dermis. Mice also have a higher pelage density compared to humans; furthermore, the permeability and fragility of the stratum corneum are higher for rodents than for humans [55,56,57]. Moreover, humans are diurnal animals, while mice are nocturnal animals, which means that mice depend mostly on vitamin D supplements for their vitamin D stores, while humans rely on UVB exposure to synthesize vitamin D [1].

Singh et al. also determined the C3-epimer, though not in all ages—only among those less than one year old [2].

## 7. C-3 Epimer Levels in Newborn and Adults

The epimer for children (<1 year) accounted for 8.7–61.1% of overall 25OHD. It was also hypothesized that the formation of epimers might depend on the maturity of the liver because the number of epimers is reduced in the blood of the infants compared to adults [2]. Moreover, van den Ouweland and colleagues established that the percentage of the C3-epimer of 25OHD was up to 60% higher in infants compared to only 22% in adults [58]. Moreover, Schleicher and colleagues noticed that adult pregnant women have higher concentrations of epimers (80% detectable epimers) compared to blood donors (43% detectable epimers). In this study, 3-epi-25OHD3 was found in all pregnant women and their cord blood, comprising 6.0% and 7.8% of 25OHD3, respectively. When the epimer was not accounted for in vitamin D estimations, 38% of women and 80% of newborns were classified as having an insufficient concentration (<75 nmol/L). However, with the epimer included in the estimation of insufficiency, 33% of women and 73% of neonates were found to have sufficient levels of vitamin D. It is assumed that a high use of dietary supplements in women could have contributed to the formation of high levels of epimers in both maternal and cord blood [3,59,60].

Vitamin D deficiency is generally regarded as 30 nmol/L or lower, while sufficient vitamin D levels for healthy people range between 40 and 80 nmol/L [7]. The range of absolute epimer concentrations for the infant population is 0–230 nmol/L, with an average of 18.2 nmol/L, while the range for epimer concentration in the adult population is around 0–22.5 nmol/L, with a mean value of 4.3 nmol/L. It is concluded that as the infants grow older, their epimers levels start decreasing. These levels remain approximately constant during their entire adult life. The percentage of C3-epimers of the total 25OHD3 ranged between 0% and 61.1% for infants and 0% and 47% for adults, with an average of 21.4% for infants and 5.9% for adults, respectively [19,61,62]. According to Chailurkit et al. (2015), the concentration of vitamin D epimer was greater in males than in females [63]. According to Keevil and colleagues, insufficient levels of vitamin D are estimated as below 50 nmol/L, while the epimer concentration could be equal to 2.5 nmol/L. Therefore, clinical interpretation is not hugely affected in adults or children [19,62]. Nonetheless, when the epimers were quantified, a percentage of infants and adults were misclassified as sufficient, with values determined at 9% and 3%, respectively [64]. Keevil et al. concluded that the epimer for 25OHD3 performs a significant role in clinical applications; however, it could have a minimal effect on routine LC–MS/MS measurements [62]. It was also shown that high levels of epimers can be found in pregnant women and their newborns, and it was suggested that the role of epimers in characterizing vitamin D status in pregnancy and infancy is imperative [3]. Others suggest that clinicians should determine the C3-epimer during routine analysis, especially among pediatric and infant populations [19]. The production of the vitamin D epimer in humans is variable [51]. Moreover, levels of 3-epi-25OHD3 production could be related to gender, age, and living areas (rural areas vs. urban areas) [62]. 

## 8. Techniques for Measurement of Vitamin D Epimers

The epimers of vitamin D have been investigated in many studies [5,6,50,61,62,64,65,66,67,68]. Immunoassay techniques have difficulty separating the D2 form of vitamin D from D3 forms, and these techniques can only give the total 25OHD value in a measurement [6,69,70,71,72]. The differentiation between 25OHD2 and 25OHD3 in most immunoassay techniques is not possible, as these techniques consider both as one entity [73]. Current LC–MS/MS techniques can separate D2 forms from D3 forms, but they have difficulty in separating the interference caused by the epimeric and isobaric metabolites. In some vitamin D LC–MS/MS with electrospray ionization techniques, the ionization potency of 3-epi-25OHD3 is higher than that of 25OHD3, which will lead to an overestimation of 3-epi-25OHD3 if it is not separated from its non-epimeric components [23]. These techniques lack specificity, which affects the estimation of major vitamin D metabolites and will lead to false-positive results [74,75]. Therefore, it is essential to separate epimers from the primary active forms of vitamin D that can overlap and compromise the accuracy of the method [5]. In the past 10 years, innovations in analytical techniques have had the beneficial ability of separating vitamin D forms from co-eluting epimers, and there are many published methods which can separate the epimers of vitamin D [5,6,49]. The separation of epimers from vitamin D metabolites is considered a challenge as these epimers have the same mass-to-charge ratios as non-epimeric metabolites. Further, mass differentiation is not possible using most standard LC–MS techniques as these epimers require chromatographic separation [19,76].

It is known that LC–MS/MS is the most recommended method for analysis of vitamin D, especially in young infants, due to its ability to detect and quantify epimeric and non-epimeric metabolites separately in plasma [23,77]. One study illustrated that ultra-performance supercritical fluid chromatography–tandem mass spectrometry (UPSFC–MS/MS) could be used as an alternative technique to ultra-high-performance liquid chromatography–tandem mass spectrometry (UHPLC–MS/MS) for the quantification of vitamin D metabolites in clinical applications. Furthermore, this technique has advantages compared to HPLC, which utilizes CO_2_ as supercritical fluid; therefore, UPSFC –MS/MS is more cost-effective than HPLC [67].

According to the literature, there are four columns widely used and tested according to their resolution, selectivity, efficiency, and analysis time in order to separate 25OHD3 from its epimer; these columns are COSMOSIL cholester (stationary phase: cholester), Kinetex F5 (stationary phase: pentafluorophenyl), Kinetex biphenyl (stationary phase: biphenyl), and COSMOSIL pentabromophenyl (stationary phase: pentabromophenyl) [68]. Cholester columns do not separate epimers of 25OHD3, while F5 and Biphenyl columns can separate the epimers. However, the analysis time for biphenyl is longer than that of the F5 column. Also, the width of the peak by biphenyl is more significant. Pentabromophenyl, on the other hand, can separate the epimers. Nonetheless, the analysis time for pentabromophenyl is longer than that for the F5 column. Also, the width of the peak in pentabromophenyl is larger. COSMOSIL pentabromophenyl could be improved if the particle size becomes smaller, as in F5, subsequently reducing the time and improving the resolution and peak shape. Lastly, the F5 column is the best column for separating 25OHD3 and its epimer [68]. According to Singh et al., a longer 5-dinitrobenzoyl-(R)-phenylglycine column (Chirex-PGLY and DNB 250 mm × 4.6 mm) gave good separation of the epimers of 25OHD. Unfortunately, the C3-epimer was not determined in all ages but only for those less than one year old [2]. Schleicher and colleagues used ultra-performance liquid chromatography–tandem mass spectrometry (UPLC–MS/MS) together with a pentafluorophenyl (PFP) column to determine the epimers [66].

## 9. Quality Assurance in C-3 Epimers Determination

Recently, the International Vitamin D External Quality Assessment Scheme (DEQAS) survey noted that liquid chromatography–tandem mass spectrometry (LC–MS/MS), immunoassays, DBP-based assays, and HPLC are commercial assays that can differentiate between 25OHD3 and its epimers. Around 14 methods were selected and compared in this article. These methods are EIA (enzyme immunoassay), Abbott Architect, chromatographic ligand binding assay, automated IDS (ImmunoDiagnostic Systems), DiaSorin RIA (radioimmunoassay), DiaSorin Liason Total immunoassay, DIAsource immunoassay, HPLC, LC–MS/MS, IDS RIA, IDS-iSYS, IDS EIA, Roche Total 25OHD, and Siemens ADVIA Centaur. Some of those methods can recognize 3-epi-25OHD3 among 25OHD3, including LC–MS/MS, HPLC, Roche Total 25OHD, DIAsource, and chromatographic ligand binding assay [19]. Gallo and colleagues determined that IDS-EIA and DiaSorin-RIA could be challenging to use for the measurement of 25OHD3 because other vitamin D metabolites could interfere with 3-epi-25OHD3 and 24,25(OH)_2_D3, which could lead to inaccurate measurements of 25OHD3 [77]. A recent study highlighted that the enzyme-linked immunosorbent assay (or a commercial chemiluminescence vitamin D assay) and the LC–MS/MS method are not equally able to evaluate vitamin D status in humans. This is because the majority of commercial assay kits cannot distinguish between 25OHD and its epimer, which leads to an overestimation of vitamin D levels as compared to LC–MS/MS, due to the interfering epimers [5].

In order to improve the accuracy and separation of 25OHD measurements in plasma, a Standard Reference Material (SRM) was developed by the National Institute of Standards and Technology (NIST) in cooperation with the National Institutes of Health’s Office of Dietary Supplements (NIH-ODS). SRM 972 contains four different concentration levels of vitamin D serum samples, and each level has different ratios of vitamin D metabolites (25OHD3, 25OHD2, 3-epi-25OHD3). The standard measurement for the SRM was carried out under three methods using isotope-dilution (ID) mass spectrometry at the Centers for Disease Control and Prevention (CDC) and the NIST [40]. The columns that were assessed in these measurements were C18 and cyano columns, and it was observed that cyano is a better column than C18 for resolving 25OHD3 and its epimers [40]. The Vitamin D External Quality Assessment Scheme (DEQAS) also uses NIST standards to ensure the analytical reliability of 25OHD, 1,25(OH)_2_D, and 3-epi-25OHD3 assays [78]. This review highlights the fact that the epimers of vitamin D should be separated from the vitamin’s non-epimeric components to properly determine circulating vitamin D levels in humans. This is especially important if one has to analyze samples from pregnant women, newborns under 1 year old, and people with liver and kidney disorders. LC–MS/MS techniques should be implemented in all the labs around the world, especially in Middle Eastern countries where these techniques are non-existent. It is also very important that the current methods in laboratories be standardized across the world to account for the interference caused by epimers and isobars. This will help to accurately address the epidemic caused by vitamin D deficiency.

## 10. Conclusions

Most of the existing methods of vitamin D analysis can lead to an overestimation of vitamin D levels due to interfering epimers, especially among infant groups and pregnant women. The determination of vitamin D metabolites and vitamin D epimers is not very reliable due to cross-reactivity issues. Studies have shown that LC–MS/MS is the gold standard method to quantify and separate epimers, bypassing the issues caused by their interference. All major vitamin D metabolites can be epimerized at the C3 position. It is known that very few labs in the world account for misleading measures due to overlapping C3-epimers. It is also known that routine vitamin D blood tests for healthy adults will not be significantly affected by epimeric interference using LC–MS/MS assays. Recent genetic models also show that the genetic determinants and potential factors of C3-epimers differ from those of non-C3-epimers.

Moreover, mice have higher epimer concentrations than humans, and the epimerization process for mice with orally ingested vitamin D3 is higher than that for mice irradiated by UVB light. It was established that oral supplementation of vitamin D could cause an increased production of epimers in mice but not in humans. Under clinical laboratory conditions, time and throughput are essential concerns. For example, separating isomers by chromatographic analysis using HPLC, LC–MS, and LC–MS/MS is time-consuming and utilizing chiral or CN columns could lead to compromised throughput. However, PFP columns could be more suitable. The biological origin of these epimers has not yet been clearly identified, and more clinical research is required [1,19,51,61,64,65,66,79,80]. The C3-epimers of vitamin D may well have an important role to play in clinical research, and more research is warranted.

## Figures and Tables

**Figure 1 ijms-21-00470-f001:**
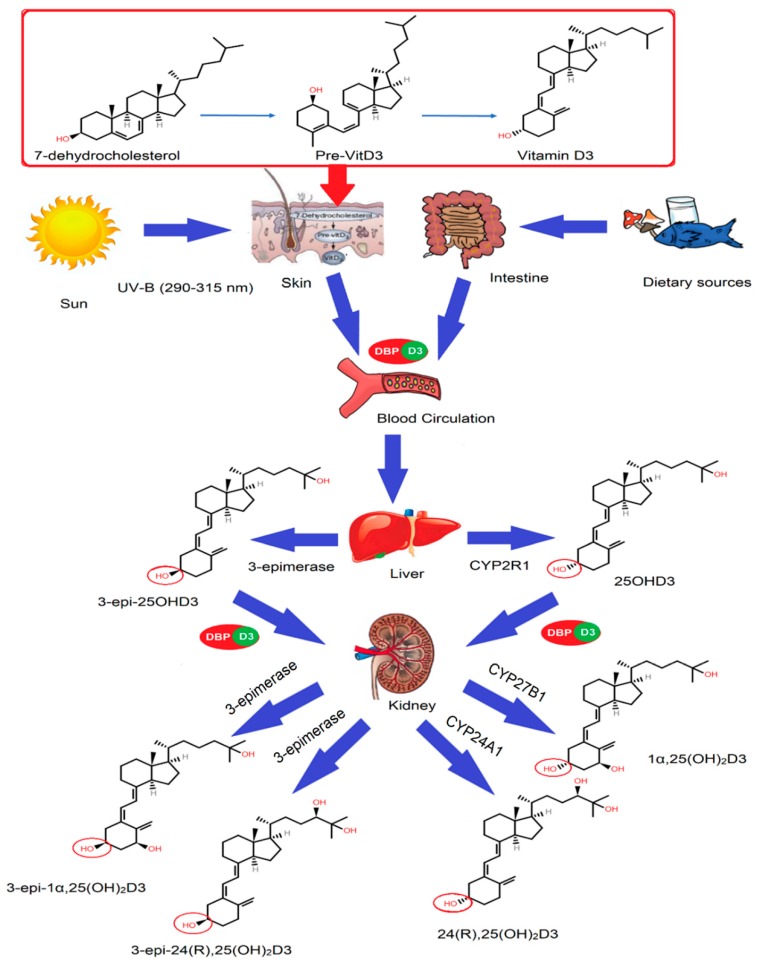
Sources and metabolism of vitamin D. Sunshine activates a chemical reaction in the plasma membrane of dermal fibroblasts and epidermal keratinocytes in the skin, producing an unstable form, 7-dehydrocholesterol, which forms pre-vitamin D3 and, upon thermal isomerization, produces a stable vitamin D3 form. Dietary sources are another source of vitamin D3 and D2. Hydroxylation of vitamin D3 occurs in the liver using enzyme CYP2R1, mainly forming 25OHD3. In the kidney, 25OHD3 undergoes further hydroxylation at the C-1α or C-24 positions. The CYP27B1 enzyme is responsible for C-1α hydroxylation, while CYP24A1 is responsible for C-24 hydroxylation. CYP24A1 can inactivate 25OHD3 to produce 24R,25(OH)_2_D3, while 3-epimerase enzyme could inactivate the major metabolites (25OHD3, 24R,25(OH)_2_D3, and 1α,25(OH)_2_D3) in the epimerization process by changing the orientation of only one group.

**Figure 2 ijms-21-00470-f002:**
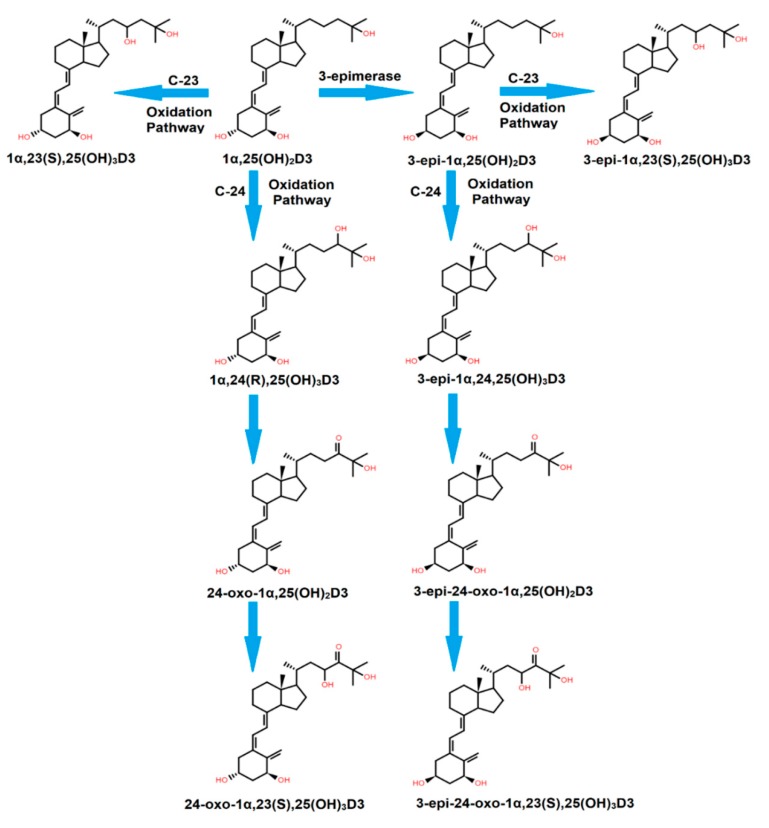
The C-23 and C-24 oxidation pathways for 1α,25(OH)_2_D3 and its epimer.

**Figure 3 ijms-21-00470-f003:**
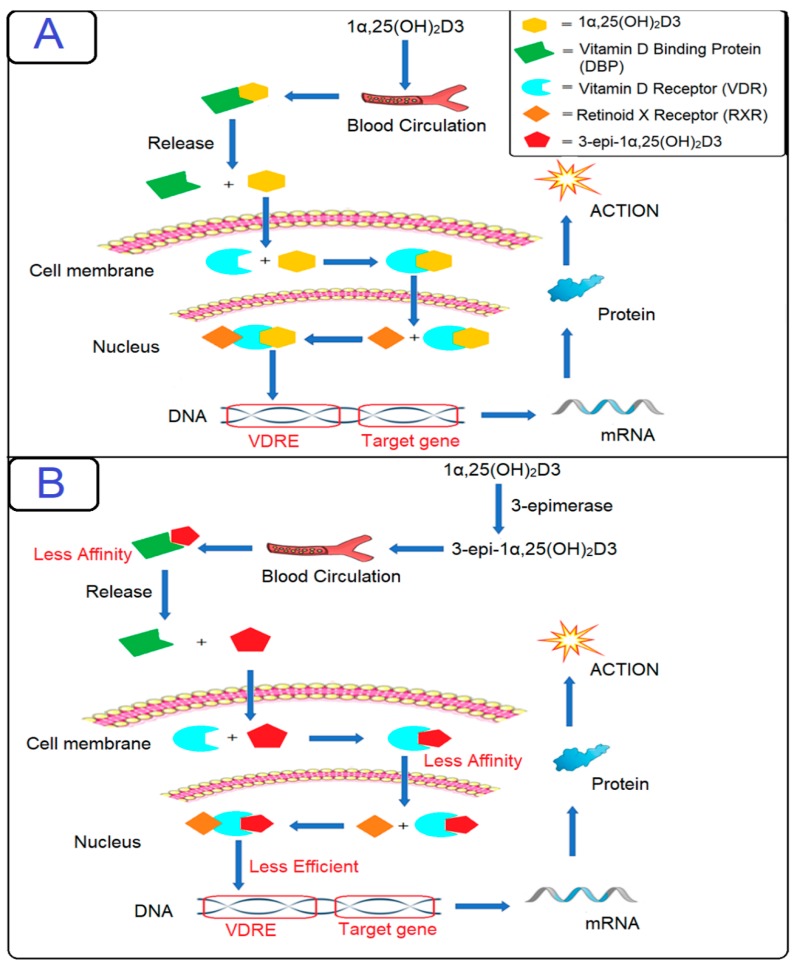
Comparison between 1α,25(OH)_2_D3 and 3-epi-1α,25(OH)_2_D3 in terms of gene regulation. (**A**) When 1α,25(OH)_2_D3 is released from DBP, it will cross the cell membrane and enter the target cell to bind with VDR. The 1α,25(OH)_2_D3–VDR complex undergoes translocation to the nucleus and performs conformational changes in order to link with other transcriptional factors and heterodimerize with RXR. After that, the 1α,25(OH)_2_D3–VDR–RXR complex binds to the vitamin D response element (VDRE); then, the transcription of RNA begins for the specific genes that are then expressed as different proteins responsible for vitamin D homeostasis. (**B**) We assume that 3-epi-1α,25(OH)_2_D3 will carry out the same function but at a slower rate. The 3-epi-1α,25(OH)_2_D3 complex could likely perform the same gene regulation as its non-epimeric forms but at a lower rate.

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
