# Peer review of "Epimers of Vitamin D: A Review"

_ijms, 2020, doi:10.3390/ijms21020470_

Round 1

Reviewer 1 Report

I would like to thank the authors for considering edits on the previous form of the manuscript. The topic is of definite interest to the field, and would be a great review of the literature. However, the paper is still very difficult to read and the readers may opt out of reading.

I kindly suggest to contact a colleague in the field or an agency to proofread the article before being published.

Author Response

Comment-1:- I would like to thank the authors for considering edits on the previous form of the manuscript. The topic is of definite interest to the field, and would be a great review of the literature. However, the paper is still very difficult to read and the readers may opt out of reading.

I kindly suggest to contact a colleague in the field or an agency to proofread the article before being published.

Ans:- Thank you for your comments. We have done a professional editing of our article from MDPI editing services from colleagues in the field and we are confident that the article reads perfectly. Please attached find the editing certificate.

Reviewer 2 Report

The authors report on the sources, formation, metabolism, function, biological activity and potency of C3-epimers of vitamin D. 

The efficiency and efficacy are not defined. What is the right definition of efficiency and efficacy? What is the scientific substrate choosing the factors affecting the epimerization of vitamin D, its efficacy and efficiency?

Some headings are not relevant because paragraphs are enormous and contain extra information. The transitions among sentences need adjusting as they create a lack of cohesion and unity. I suggest the text to be organized into more paragraphs with appropriate headings and transitions among sentences. Sentences are complex, and I had to read several times to understand the idea that the authors promote (line 100-105).

I included some examples that attracted my attention.

Introduction:

Epimers importance? I would start the introduction by highlighting the importance of Vit D epimers. Alternatively, they focus mostly on vit D. Moreover, they undermine the role of the epimers: “The role of epimers in disease onset and progression is not clearly understood but it is known that epimers are less potent in their role as compared to non-epimeric forms” (line 74-75).

Line 77-80:

There are two primary metabolites of vitamin D, vitamin D3, and vitamin D2, collectively known as vitamin D. Vitamin D3 in the skin is generated in the skin (skin repeated) by the conversion of 7- 79 dehydrocholesterol to pre-vitamin D3 followed by vitamin D3 (not clear), as shown in Figure 1 [13–15]. The second step is… (there is no mention which is step one?).

Line 82:

These metabolites in the kidneys get converted. (into what? not complete sentence)

Line 132:

Furthermore, the epimerase enzymes can metabolize the epimerization process… (is this correct? Really?)

Line 249-250:

Vitamin D epimer levels are the heading of a paragraph, however, it starts with vit D deficiency, which in my opinion is irrelevant.

Line 412-416:

Here is one example of incorrect use of transition Moreover.

Most of the existing methods of vitamin D analysis could lead to an overestimation of vitamin D levels due to interfering epimers, especially in infant groups and pregnant ladies. The determination of vitamin D metabolites and its epimers are not very reliable due to the cross-reactivity issues. Moreover, LC-MS/MS is regarded as a gold standard method to quantitate and separate the interferences caused by epimers. (Moreover is not correctly used here, it is not additional information to previous sentences, but a piece of new information).

Author Response

Reviewer 2

Comment-2:- The authors report on the sources, formation, metabolism, function, biological activity and potency of C3-epimers of vitamin D. 

The efficiency and efficacy are not defined. What is the right definition of efficiency and efficacy? What is the scientific substrate choosing the factors affecting the epimerization of vitamin D, its efficacy and efficiency?

Ans:- Thank you. We have removed these words from text and reworded the sentence as follows.

“C3-epimerization of vitamin D”.

Comment-3:- Some headings are not relevant because paragraphs are enormous and contain extra information. The transitions among sentences need adjusting as they create a lack of cohesion and unity. I suggest the text to be organized into more paragraphs with appropriate headings and transitions among sentences.

Ans:- Thank you for these comments. We have adjusted the transition among sentences. We have also organized text into more paragraphs with appropriate headings.

Comment-4:- Sentences are complex, and I had to read several times to understand the idea that the authors promote (line 100-105).

Ans:- We have corrected all the text and the following paragraph line (100-105).

“In neonatal human keratinocytes, Reddy and colleagues have concluded that 3-epi-1α,25(OH)2D3, is metabolized through the C-24 oxidation pathways to produce three polar compounds: 3-epi-1α,24,25(OH)3D3, 3-epi-24-oxo-1α,25(OH)2D3, and 3-epi-24-oxo-1α,23(S),25(OH)3D3, while C-23 oxidation of 3-epi-1α,25(OH)2D3 produces 3-epi-1α,23(S),25(OH)3D3, as shown in Figure 2 [13,19]”.

We have also done a professional editing of our article from MDPI editing services from colleagues in the field and we are confident that the article reads perfectly now. We have introduced more headings in our large paragraphs and we have smoothed out the transition between sentences and they reads perfect now.

Comment-5:- I included some examples that attracted my attention.

Introduction:

Epimers importance? I would start the introduction by highlighting the importance of Vit D epimers. Alternatively, they focus mostly on vit D.

Ans:- We have included a paragraph on the importance of epimers in the beginning of introduction as follows.

“Recently it was found that C3-epimers of vitamin D may have an important role to play in clinical research. The accurate source of C3-epimers is not known but It was found that the oral supplementation of vitamin D can cause increased production of epimers in mice but not in humans. Moreover, it is known that all major vitamin D metabolites can be epimerized at the C3 position, with higher amounts in infants. Furthermore, mothers and newborns are known to have high levels of C3-epimers. Recent genetic models also show that the genetic determinants and potential factors of C3-epimers differ from those of non-C3-epimers. It is also of importance to know that the C3-epimers can cause an overestimation of vitamin D status in routine laboratory tests. However, very few labs in the world take into account that measurements can be misleading due to overlapping C3-epimers and that these co-eluting C3-epimers could be separated from vitamin D metabolites using LC-MS/MS techniques. However, studies have shown that routine vitamin D blood tests for healthy adults are not significantly affected by epimeric interference using LC-MS/MS assays”.

Comment-6:- Moreover, they undermine the role of the epimers: “The role of epimers in disease onset and progression is not clearly understood but it is known that epimers are less potent in their role as compared to non-epimeric forms” (line 74-75).

Ans:- Thank you. We have reworded the sentence as follows

 “The role of epimers in disease onset and progression is not clearly understood, but it is known that epimers have a potent role as compared to their corresponding non-epimeric forms [2,3,8–11]”.

Comment-7:- Line 77-80:

There are two primary metabolites of vitamin D, vitamin D3, and vitamin D2, collectively known as vitamin D. Vitamin D3 in the skin is generated in the skin (skin repeated) by the conversion of 7- 79 dehydrocholesterol to pre-vitamin D3 followed by vitamin D3 (not clear), as shown in Figure 1 [13–15]. The second step is… (there is no mention which is step one?).

Ans:- We have revised the above paragraph as follows and corrected and highlighted it in the text

Vitamin D3 is formed from its precursor 7-dehydrocholesterol in the skin by ultraviolet B light (medium wavelength, 290-315 nm). In the first step, 7-dehydrocholesterol is converted to pre-vitamin D3 which is followed by conversion of pre-vitamin D3 to vitamin D3, as shown in Figure 1 [13–15]. The second step is governed by the conversion of vitamin D3 to 25OHD3 in the liver via the 25-hydroxylase (CYP2R1) enzyme, as shown in Figure 1 [16]”.

Comment-8:- Line 82:

These metabolites in the kidneys get converted. (into what? not complete sentence)

Ans:- Thank you. We have made this sentence more clear and corrected and highlighted it in the text, see as follows

25OHD3 and 3-epi-25OHD3 are converted in the kidneys into 1α,25(OH)2D3 via the action of the enzyme 1α-hydroxylase (CYP27B1), as shown in Figure 1 [17,18];

Comment-9:- Line 132:

Furthermore, the epimerase enzymes can metabolize the epimerization process… (is this correct? Really?)

Ans:- Furthermore, epimerase enzymes can carry out the epimerization process of 1α,25(OH)2D3 and 24(R)25(OH)2D3 but not at the same rate as 25OHD3. Moreover, the observation shows that this process is irreversible [17,25].

Comment-10:- Line 249-250:

Vitamin D epimer levels are the heading of a paragraph, however, it starts with vit D deficiency, which in my opinion is irrelevant.

Ans:- We have removed the following paragraph from under this heading.

“Vitamin D deficiency is generally regarded as 30 nmol/L or lower, while sufficient vitamin D levels for healthy people range between 40 and 80 nmol/L”.

Comment-11:- Line 412-416:

Here is one example of incorrect use of transition Moreover.

Most of the existing methods of vitamin D analysis could lead to an overestimation of vitamin D levels due to interfering epimers, especially in infant groups and pregnant ladies. The determination of vitamin D metabolites and its epimers are not very reliable due to the cross-reactivity issues. Moreover, LC-MS/MS is regarded as a gold standard method to quantitate and separate the interferences caused by epimers. (Moreover is not correctly used here, it is not additional information to previous sentences, but a piece of new information).

Ans:- Thank you. We have corrected the above paragraph as follows.

Most of the existing methods of vitamin D analysis could lead to an overestimation of vitamin D levels due to interfering epimers, especially in infant groups and pregnant ladies. The determination of vitamin D metabolites and vitamin D epimers is not very reliable due to cross-reactivity issues. Studies have shown that LC–MS/MS is the gold standard method to quantify and separate epimers, bypassing the issues caused by their interference.

Round 2

Reviewer 1 Report

The manuscript reads well and is close to publication ready. I recommend the manuscript for publication with a few small suggestion:

As written now the Highlights section lists a number of seemingly random information points and/or point addressed by the manuscript. I suggest condensing the highlights to the most important point the reader should take away after reading the paper:

Highlights

All major vitamin D forms can undergo epimerization at C3 position, leading to the formation of C3-epimers that can overestimate vitamin D status in routine laboratory tests. Higher levels of C3-epimers are observed in mothers and newborns. LC-MS/MS can be used to separate C3-epimers from other vitamin D metabolites

Also Pg.2 – Last lane – I would remove the word However due to the repetitive use…. Instead of – However, studies, … should be Studies…..

Author Response

Answer to Reviewer comments

Question-1:- The manuscript reads well and is close to publication ready. I recommend the manuscript for publication with a few small suggestion:

Ans: - Thank you.

Question-2 :- As written now the Highlights section lists a number of seemingly random information points and/or point addressed by the manuscript. I suggest condensing the highlights to the most important point the reader should take away after reading the paper:

Ans: - Thank you. We have condensed the highlights section and it reads as follows

Question-3:- Highlights

All major vitamin D forms can undergo epimerization at C3 position, leading to the formation of C3-epimers that can overestimate vitamin D status in routine laboratory tests.

Higher levels of C3-epimers are observed in mothers and newborns.

LC-MS/MS can be used to separate C3-epimers from other vitamin D metabolites

Ans: - Thank you. We have condensed the highlights section as per the reviewer suggestion and it reads as follows. This is highlighted in text as green.

Highlights

All major vitamin D forms can undergo epimerization at C3 position, leading to the formation of C3-epimers that can overestimate vitamin D status in routine laboratory tests. Higher levels of C3-epimers are observed in mothers and newborns. Oral supplementation of vitamin D can cause an increased production of epimers in mice. LC-MS/MS can be used to separate C3-epimers from other vitamin D metabolites.

Question-4:- Also Pg.2 – Last lane – I would remove the word However due to the repetitive use…. Instead of – However, studies, … should be Studies…..

Ans: - Thank you. We have removed the word however and it reads as follows. This is highlighted in in text in green.

Studies have shown that routine vitamin D blood tests for healthy adults are not significantly affected by epimeric interference using LC-MS/MS assays.

Reviewer 2 Report

The authors fulfilled the requests of the reviewers properly.

Author Response

Many thanks

This manuscript is a resubmission of an earlier submission. The following is a list of the peer review reports and author responses from that submission.

Round 1

Reviewer 1 Report

This review article describes potentially interesting topic to vitamin D researchers, i.e. vitamin D nutritional status can be inaccurately defined due to measurement techniques that do not distinguish epimersepimers from major metabolites of vitamin D. However, manuscript suffers from poor organization and lack critical synthesis of information needed for review article.

Redundancy: some examples are below p4. paragraph 1 and 2  p5. section 2 natural synthesis of vitamin D.... information in first two paragraphs are already introduced in section 1. Subtitles do not reflect the content.  For example section 1.1 is not about causes of vitamin D deficiency Misinterpretation of literature or unclear expression of authors' intent.  For example section 3. the biological activity of vitamin D epimer.... describes interrelationship between vitamin D, PTH, calcitonin and serum calcium as well as role of vitamin D in gene regulations.  The relationships among the nutrients and hormones are not accurately written, and role of active vitamin D metabolite in the gene regulation is not correctly described.   Grammatical errors:  Run-on sentenses Inappropriate use of semi-colons and commas

It would have been much better if the authors focused on why current vitamin D measurement techniques are inadequate for determining true vitamin D nutritional status and what is known about epimer formation rather than trying to cover vitamin D metabolism as well as its function.  Latter part could have been much more brief, just to show how they relate to epimer formation and interpretation of serum vitamin D values.  

Reviewer 2 Report

The review is interesting and addresses an important topic: Vitamin D epimers, from production to effect on estimation of total vitamin D. The overall structure of the review is good and well organized however, extensive edits will be required to make the content easily understood by the readers.

Highlights is a mixture of items to be addressed and learning points – I suggest the authors stick to one format., ideally short take home messages that the reader could take away without reading the whole paper.

Minor comments (only a short list of the identified)

“Sources that effects C3-epimers levels in mice” – Sources that affect C3-epimers…(pg1)

“C3-epimers can causes” – C3 epimers can cause...(pg1)

Gallo and collegues find out (found out, identified, etc) ... (pg.16).